# Protective Efficacy of a Novel DNA Vaccine with a CL264 Molecular Adjuvant against *Toxoplasma gondii* in a Murine Model

**DOI:** 10.3390/vaccines12060577

**Published:** 2024-05-25

**Authors:** Kunping Ju, Yunnan Zhang, Zhaolin Xu, Lingyu Li, Xiaoyan Zhao, Huaiyu Zhou

**Affiliations:** Department of Pathogen Biology, School of Basic Medical Sciences, Cheeloo College of Medicine, Shandong University, Jinan 250012, China; 201900412088@mail.sdu.edu.cn (K.J.); 15621898329@163.com (Y.Z.); 15864237509@163.com (Z.X.); lingyu_li2022@163.com (L.L.); 15736031707@163.com (X.Z.)

**Keywords:** *Toxoplasma gondii*, IST, DNA vaccine, immunity, adjuvant

## Abstract

Toxoplasmosis is a significant global zoonosis with devastating impacts, and an effective vaccine against toxoplasmosis for humans has not yet been developed. In this study, we designed and formulated a novel DNA vaccine encoding the inhibitor of STAT1 transcriptional activity (IST) of *T. gondii* utilizing the eukaryotic expression vector pEGFP-N1 for the first time, with CL264 being a molecular adjuvant. Following intramuscular injection of the vaccine into mice, the levels of antibodies and cytokines were assessed to evaluate the immune response. Additionally, mice were challenged with highly virulent RH-strain tachyzoites of *T. gondii*, and their survival time was observed. The results show that the levels of IgG in serum, the ratio of IgG2a/IgG1 and the levels of IFN-γ in splenocytes of mice were significantly higher in the pEGFP-TgIST group and the pEGFP-TgIST + CL264 group than in the control group. In addition, the proportion of CD4+/CD8+ T cells was higher in mice immunized with either the pEGFP-TgIST group (*p* < 0.001) or the pEGFP-TgIST + CL264 group (*p* < 0.05) compared to the three control groups. Notably, TgIST-immunized mice exhibited prolonged survival times after *T. gondii* RH strain infection (*p* < 0.05). Our findings collectively demonstrate that the TgIST DNA vaccine elicits a significant humoral and cellular immune response and offers partial protection against acute *T. gondii* infection in the immunized mice, which suggests that TgIST holds potential as a candidate for further development as a DNA vaccine.

## 1. Introduction

*Toxoplasma gondii*, an obligatory intracellular opportunistic protozoon with worldwide distribution, can cause toxoplasmosis [1,2,3]. While cats are the definitive hosts for *T. gondii*, nearly all homeothermic animals, including humans, can become infected [4]. Human infection is primarily acquired through the consumption of food or water contaminated with oocysts shed from cats, or by eating raw or undercooked meat containing tissue cysts [5,6]. Although most infections are asymptomatic, immunocompromised individuals may experience severe consequences, including encephalitis, myocarditis, or pneumonitis [7]. Additionally, congenital toxoplasmosis can lead to severe fetal malformations, abortion and perinatal death [4].

Given the grave consequences of toxoplasmosis, effective treatment or prevention measures are essential. Despite some drugs being used to control acute *T. gondii* infections, no available drugs can cure persistent infections [8]. Thus, vaccination offers a promising alternative to combat toxoplasmosis [4,8]. Notwithstanding the Toxovax™ vaccine, composed of the S48 mutant live strain, which is the only licensed vaccine for preventing abortion in sheep [4,9], an effective vaccine against toxoplasmosis for humans remains an urgent need [8].

Immunogenicity plays a crucial role in vaccine development, and several virulence factors of *T. gondii*, such as surface antigens (SAGs), micronemal (MIC) proteins, dense granule antigens (GRAs), and rhoptry proteins (ROPs), have been considered as potential vaccine targets [10]. However, despite using various antigens in vaccine development against *T. gondii*, no effective vaccines have been identified [11].

One promising candidate is the *T. gondii* inhibitor of STAT1 transcriptional activity (TgIST), a protein secreted by intracellular parasites that can inhibit STAT1-dependent proinflammatory gene expression by trafficking to the host cell nucleus [12]. *T. gondii* blocks STAT1-mediated gene expression by interacting with phosphorylated STAT1 dimers, leading to unexpected transcriptional silencing [13]. A study has shown that TgIST-deficient parasites are rapidly cleared by Gr1+ inflammatory monocytes in mouse models, highlighting the protective role of TgIST against IFN-γ-mediated killing [12]. TgIST comprises a core region with internal repeats, which forms a helical conformation upon binding to the STAT1 dimers and is necessary and sufficient to block STAT1-mediated gene expression [13]. Understanding the importance of TgIST in toxoplasmosis and its underlying mechanisms could provide new avenues for controlling *T. gondii* infection by targeting TgIST.

In addition, adjuvants play a significant role in most clinical vaccines as they enhance the induction of adaptive immunity by the active components of vaccines [14,15]. Toll-like receptor 7 (TLR7) is predominantly expressed in plasmacytoid dendritic cells (pDCs), macrophages and B cells [16]. TLR7 modulators have shown promise in various applications, including as interferon inducers, anti-cancer agents, anti-infectious agents, therapeutic agents for immunological diseases and vaccine adjuvants [17,18]. TLR-dependent mechanisms are crucial in parasite recognition and in inducing IFN-γ production by NK cells during *T. gondii* infection [19]. Among these, TLR7 ligands hold particular promise as they induce prominent Th1-type immune responses [20]. Thus, we select CL264 (a 9-benzyl-8 hydroxyadenine derivative containing a glycine on the benzyl group), a TLR7 agonist, as an adjuvant for our investigation.

In the present study, we evaluated the immunogenicity and protective efficacy of the pEGFP-TgIST DNA vaccine using CL264 as an adjuvant in a BALB/c mouse model. Also, we analyzed the ability of the DNA vaccine in protecting against *T. gondii* infection with the highly virulent RH strain.

## 2. Materials and Methods

### 2.1. Mouse and Parasite

Female BALB/c mice aged 6–8 weeks were purchased from Jinan Pengyue Experimental Animal Breeding Limited Company (Jinan, China) and used for the vaccination study. The RH strain of *T. gondii* was maintained at −80 °C in our laboratory and restored to high virulence by peritoneal passage in BALB/c mice for use in lethal challenge experiments. All the mice were provided with *ad libitum* access to food and tap water and were handled following good animal practice guidelines and animal ethics procedures.

### 2.2. Construction of Recombinant pEGFP-TgIST

Primers were designed based on the TgIST gene (ToxoDB: TGRH88_087790, available on 1 February 2022 https://toxodb.org/toxo/app/record/gene/TGRH88_087790#Sequences) and pEGFP-N1 plasmid (backbone), as shown in Appendix A. The target gene TgIST and the backbone were amplified by PCR using the corresponding primers. The PCR products of the TgIST gene were then ligated into the pEGFP-N1 plasmid (Invitrogen, Waltham, MA, USA) through homologous recombination to construct the recombinant plasmid pEGFP-TgIST. In brief, a 10 μL homologous recombination system containing the target gene TgIST, linearized pEGFP-N1 plasmid, double-distilled water (ddH_2_O) and 2×Clon Express Mix was prepared following the instructions, reacted at 50 °C for 5 min, and then immediately placed at 4 °C. The recombinant plasmids were identified by PCR, *BamH* I and *Nde* I double restriction enzyme digestion and DNA sequencing.

The positive plasmids were purified from transformed *Escherichia coli* DH5α cells following the manufacturer’s instructions (EndoFree plasmid Giga kit, Qiagen Sciences, Germantown, MD, USA). The concentration of the extracted recombinant plasmids was determined by spectrophotometry at OD260 and OD280 and diluted in sterile phosphate-buffered saline (PBS) to a concentration of 1 µg/µL.

### 2.3. Expression of TgIST In Vitro

HEK 293T cells were cultured in Dulbecco’s modified Eagle’s medium (DMEM) supplemented with 100 mg/mL streptomycin/penicillin and 10% fetal bovine serum (FBS) at 37 °C and 5% CO_2_ in a six-well plate before transfection. The recombinant plasmid pEGFP-TgIST was then transfected into HEK 293T cells using Lipofectamine^®^ 2000 reagent (Invitrogen, Waltham, MA, USA) following the manufacturer’s instructions.

After 24 h of transfection, the expression of pEGFP-TgIST was examined under a fluorescence microscope. Untransfected HEK 293T cells (PBS-treated group) were used as the negative control. After 72 h of transfection, the expression of pEGFP-TgIST was verified by Western blot. In brief, cell lysates were prepared by adding 100 μL of cocktail and RIPA lysate to each well, followed by incubation on ice for 30 min and centrifuging at 4 °C for another 30 min. The supernatant was collected for protein extraction, and a series of protein solutions with known concentrations (ranging from 50 to 500 mg/mL) were prepared as standard references. The protein samples for Western blot were prepared to ensure equal protein amounts in each group. The extracted proteins were separated by 12% sodium dodecyl sulfate polyacrylamide gel electrophoresis (SDS-PAGE) and transferred to a polyvinylidene fluoride (PVDF) membrane. Proteins attached to the membrane were blocked with 0.5% skim milk powder, followed by incubation with a 1:500 dilution of anti-*Toxoplasma* mouse sera as the primary antibody. Sheep anti-mouse IgG antibody (1:1000 dilution) was used as the secondary antibody. Blots were visualized using an alkaline phosphatase imaging buffer containing 5-bromo-4-chloro-3-indolyl phosphate (BCIP) and nitroblue tetrazole (NBT).

### 2.4. Immunization and Challenge

In total, 100 female BALB/c mice were randomly divided into five groups (Table 1) of 20 mice each and vaccinated three times at 2-week intervals with intramuscular (*i.m*.) injection into the anterior tibialis muscle with pEGFP-TgIST or pEGFP-TgIST + CL264. Control groups received PBS, pEGFP-N1 or CL264 only by *i.m*. injection using the same schedule.

Tail bleeding was performed before immunization and 2 weeks after each immunization. The collected blood was left overnight at 4 °C and the sera were separated by centrifugation at 8000 rpm for 5 min and stored at −20 °C until use. Two weeks after the third immunization, 10 mice in each group were intraperitoneally inoculated with 100 tachyzoites of *T. gondii* RH strain per mouse, and the survival times were monitored daily and recorded. The entire vaccine preparation process is illustrated in the flowchart in Figure 1A.

### 2.5. IgG Antibody Assay

Total IgG, IgG1 and IgG2a antibodies against *T. gondii* were detected using an enzyme-linkedimmunosorbent assay (ELISA) with *Toxoplasma*-soluble antigen (STAg). The microplates were coated with 2 μg STAg per well overnight at 4 °C. After that, mouse serum samples were tested at room temperature for one hour, and reactions were performed using peroxidase-conjugated anti-mouse IgG, IgG1, or IgG2a (1:4000 dilution, Southern Biotech, Birmingham, AL, USA), respectively, as probes for 30 min at room temperature. The substrate for imaging was TMB, and the reaction was stopped by adding 2 N sulfuric acid. The absorbance at 450 nm was measured using an automated microplate reader (Thermo Scientific, Waltham, MA, USA). Serum samples were considered positive if their absorbance (AV) values were higher than the mean AV (+) 2 × standard deviation (SD) of negative control serum samples, with the pre-inoculation serum used as a negative control serum.

### 2.6. Flow Cytometry

Two weeks after the last immunization, three mice from each group were euthanized, and their spleens were aseptically harvested. Splenocytes were obtained by pushing the spleen through a nylon sieve, and then red blood cells were removed using erythrocyte lysis buffer (Solarbio, Shanghai, China). The purified splenocytes were resuspended in RPMI 1640 medium supplemented with penicillin/streptomycin (p/s) and 10% fetal bovine serum (FBS). The splenocyte suspensions were stained with phycoerythrin (PE)-labeled anti-CD3 (BioLegend), allophycocyanin (APC)-labeled anti-CD4 (BioLegend) and fluorescein isothiocyanate (FITC)-labeled anti-CD8a (BioLegend) at 4 °C for 30 min in the dark. The stained cells were then analyzed using Beckman Coulter by CytExpert 2.4.0.28 software.

### 2.7. Cytokine Assays

Splenocytes were harvested from five mice in each group two weeks after the last immunization, and were separately cultured with STAg at 37 °C in 5% CO_2_. After 72 h of incubation, cell-free supernatants were collected and assayed for IFN-γ, IL-4 and IL-10 using ELISA kits according to the manufacturer’s protocol (ABclonal, Woburn, MA, USA). A standard curve was drawn based on the diluted concentration of standard samples provided in the kit and the measured OD value. The measured OD value of experimental mice samples was converted into the concentration of cytokines (IFN-γ, IL-4 and IL-10) using the provided formula.

### 2.8. Statistical Analysis

All statistical analyses were performed using GraphPad Prism 5.0 and SPSS17.0 Data Editor (SPSS, Inc., Chicago, IL, USA). The statistical significance of the comparison of antibody levels, cytokine production, and the percentage of CD4^+^ and CD8^+^ T cells between all the groups was assessed initially with Shapiro–Wilk tests, followed by one-way analysis of variance (ANOVA). Survival curves were generated using the Kaplan–Meier method, and statistical comparisons were performed using the log-rank method.

## 3. Results

### 3.1. Identification of Recombinant pEGFP-TgIST

The target TgIST gene was successfully ligated into the eukaryotic expression plasmid pEGFP-N1 through homologous recombination, resulting in the construction of the recombinant plasmid pEGFP-TgIST. As shown in Figure 1, the TgIST gene was specific, approximate 3000 bp in length on agarose gel, and was characterized by restriction endonuclease digestion as well as PCR amplification (Figure 1B). Additionally, DNA sequencing analysis showed that the TgIST gene in pEGFP-TgIST was approximately 3000 bp with nearly 100% identity to the *T. gondii* RH strain (Appendix A), confirming the accurate construction of the recombinant plasmid.

### 3.2. Identification of pEGFP-TgIST Expression In Vitro

After 24 h of transfection, the expression of pEGFP-TgIST was examined under a fluorescence microscope by evaluating the intrinsic EGFP gene on the plasmid, which indirectly indicated the expression of TgIST fused with EGFP. Specific green fluorescence was observed in the HEK 293T cells transfected with pEGFP-N1 and pEGFP-TgIST plasmids, while no fluorescence was observed in the PBS-treated cells (Figure 2A–C). Furthermore, Western blot analysis confirmed the expression of TgIST in HEK 293T cells, with the detection of a band at 130 kDa (lane 3), while the control cells transfected with the PBS or empty plasmid did not show any band (lane 1, 2) upon incubation with the same antibody (Figure 2D).

### 3.3. Detection of IgG and Subtype IgG1, IgG2a

To assess the *T. gondii*-specific antibody response, sera from all mice were collected, and the total IgG and IgG subtypes (IgG1 and IgG2a) were analyzed using ELISA. The results are shown in Figure 3.

Significantly higher levels of IgG were observed in the sera from the pEGFP-TgIST group and pEGFP-TgIST + CL264 group compared with control groups (*p* < 0.05), and the IgG levels gradually increased and reached their peak at 4 weeks after the final immunization. However, no significant difference was observed among three control groups at any point (Figure 3A).

To further determine the IgG subtypes (IgG1 and IgG2a), mouse serum was collected at 4 weeks after the final immunization. As shown in Figure 3B, the level of IgG2a in immunized mice was significantly higher than IgG1, especially in the pEGFP-TgIST + CL264 group, which exhibited the highest IgG2a/IgG1 ratio. Conversely, IgG1 titers did not significantly differ among five groups, and there was no significant difference in IgG2a/IgG1 ratio in control groups. The results indicate that a Th1 response was dominantly induced.

### 3.4. Flow Cytometry Analysis of CD4^+^ and CD8^+^ T Cell Levels

Flow cytometry analysis was conducted to determine the T cell subsets of mice from each group. As shown in Figure 4, the percentage of CD3^+^ CD4^-^ CD8^+^ T cells in mice immunized with pEGFP-TgIST was higher than that in the PBS, CL264, or pEGFP-N1 group (*p* < 0.05). Additionally, the pEGFP-TgIST + CL264 group showed a higher percentage of CD3+ CD4^-^ CD8^+^ T cells than the controls. However, the percentages of CD3^+^ CD4^+^ CD8^-^ T cells in all mice groups did not show statistically significant differences. Moreover, the ratio of CD4^+^/CD8^+^ T cells was at a higher level in the group of mice immunized with pEGFP-TgIST (*p* < 0.001) or pEGFP-TgIST + CL264 (*p* < 0.05) compared to the three control groups.

### 3.5. Detection of Cytokines (IL-4, IL-10 and IFN-γ)

To assess the immunological response induced by DNA vaccination, the levels of cytokines, including IFN-γ, IL-4, and IL-10, in the spleen cell suspensions of immunized mice were measured using ELISA. As shown in Table 2, the levels of IFN-γ in splenocytes from mice immunized with pEGFP-TgIST (*p* < 0.05) and pEGFP-TgIST + CL264 (*p* < 0.01) were significantly higher than those in mice from the control groups; however, the levels of IL-4 and IL-10 were not significantly different in mice from each groups (Table 2).

### 3.6. Protective Efficacy Elicited by pEGFP-TgIST Vaccination

To further evaluate whether the DNA vaccine could induce protective efficacy against acute *T. gondii* infection, mice of all the groups were challenged with 100 tachyzoites of the highly virulent *T. gondii* RH strain 2 weeks after the final immunization. The survival percentages of the different groups of mice are shown in Figure 5. Significant protection was demonstrated in mice immunized with pEGFP-TgIST or pEGFP-TgIST + CL264 compared with the control groups (*p* < 0.05). Moreover, the survival percentage of mice immunized with pEGFP-TgIST + CL264 was slightly higher than that of mice immunized with pEGFP-TgIST, but the difference was not statistically significant. All the mice in the control groups died within 12 days after challenge, while 20% of the mice immunized with pEGFP-TgIST + CL264 remained alive at 20 days post-infection.

## 4. Discussion

Toxoplasmosis is a widespread parasitic disease of significant public health concern, emphasizing the need to develop an effective vaccine against the causative agent, *T. gondii* [8]. In the present study, we constructed a novel DNA vaccine, pEGFP-TgIST, against *T. gondii*, and investigated its immunogenicity and protective efficacy in the BALB/c mice. The results demonstrate that the pEGFP-TgIST DNA vaccine elicited a robust immune response and effective protection against acute *Toxoplasma* RH strain infection, suggesting that TgIST was a promising candidate for the development of a vaccine against toxoplasmosis.

IST is a member of the GRA family that can enter the host cell nucleus and inhibit the STAT1 signaling pathway, thereby reducing the production of proinflammatory factors, primarily including IFN-γ. By altering the immune status of host cells and the intracellular environment, IST facilitates immune evasion and long-term parasitism [21,22]. Previous studies have shown that *T. gondii* releases GRA to modify nanophase vesicles, helping the parasite in obtaining nutrients and evading immune responses [23]. Thus, in this study, TgIST appears to be an attractive vaccine candidate capable of enhancing both cellular and humoral immunity and improving the protective efficacy against *T. gondii* challenges.

In the quest for effective *T. gondii* vaccines, three main categories have been explored: inactivated vaccines and live attenuated vaccines, genetically engineered vaccines, and novel vaccines. Inactivated and live attenuated vaccines have shown limited immune protection, leading to reduced focus on these approaches [24,25]. As to novel vaccines, insufficient research time and shallow research degree limit its capacity to be the optimal mode of *T. gondii* vaccine. The development of novel vaccines has been limited by insufficient research time and depth. Among the genetic engineering vaccines, DNA vaccines have stood out due to their advantages, such as a simple production process, low cost, stable properties, durable immunity, ability to express configuration-related antigens, induction of cytotoxic T lymphocyte (CTL) response, simple and standardized vaccination procedures, and convenient combined immunization [26]. DNA vaccines have played a protective role in epidemic diseases including the COVID-19 epidemics against SARS-CoV and SARS-CoV-2 variants [27,28], and have even been used to potentially prevent or modify the course of neurological diseases [29]. Additionally, significant progress has been made in DNA vaccines against *T. gondii* [30].

To evaluate the potential protective immunity of pEGFP-TgIST vaccination, we assessed the levels of T cell subsets, cytokines, and antibodies in immunized mice. The increased CD4^+^ T cells and CD4^+^/CD8^+^ T cell ratio observed in the pEGFP-TgIST immunized mice indicated robust T cell-mediated immune responses, which are crucial for prolonged survival. Activated macrophages, as well as CD4^+^ and CD8^+^ T cells, play vital roles in controlling acute *T. gondii* infection and maintaining chronic infection [31]. CD4^+^ T cells are essential for the suppression of intracellular pathogens in the early stage of infection and play critical roles in adaptive immune responses [32], while CD8^+^ T cells participate in resistance through cytolytic activity in the later stage of intracellular infection [33].

The Th1-type immune response, characterized by the secretion of interleukin-12 (IL-12) and the induction of IFN-γ, is another critical path against *T. gondii* infection [34,35]. IFN-γ can promote the activation of NK cells and dendritic cells [36], and induce CD8^+^ T cells into maturing into cytotoxic T cells, which indirectly resist *T. gondii* invasion and eliminate intracellular parasites [37]. Additionally, IFN-γ induces indoleamine 2, 3-dioxygenase (IDO), which converts tryptophan to L-formylurine, leading to “tryptophan starvation” and inhibition of *T. gondii* growth in the basal metabolism [38,39].

Studies using IFN-γ receptor or STAT1-deficient mice have confirmed the vital role of IFN-γ in immunity against *T. gondii*, as these mice showed extreme susceptibility to nonlethal *T. gondii* strains [40,41,42]. In our study, significantly increased IFN-γ levels were detected in mice immunized with pEGFP-TgIST, indicating the successful induction of a Th1-type immune response. However, IL-4 and IL-10, associated with a Th2-type immune response, did not show significant changes among the vaccine groups, which indicated a more pronounced Th1 response than a Th2 response [43].

Specific antibody production is crucial for inhibiting the adherence of parasites to host cell receptors and promoting macrophages to kill intracellular parasites [44]. In this study, high levels of specific anti-*T. gondii* IgG were detected in the sera of pEGFP-TgIST immunized mice. Furthermore, IgG subclass analysis revealed a predominance of IgG2a over IgG1, confirming the induction of a Th1-type immune response by the pEGFP-TgIST vaccine.

To enhance the immune response of the vaccine, we selected CL264, a TLR7 agonist, as an adjuvant. The use of adjuvants can improve the body’s immune response, reduce vaccine dosages and save costs. Traditional adjuvants, such as aluminum salt adjuvants, can only induce humoral immune responses, with limited effects on cellular and cytotoxic T cell responses, and their quality is difficult to control. However, new adjuvants have emerged, and they can be categorized into three types: immunomodulatory molecular adjuvants, antigen delivery system adjuvants, and combinations of the first two types [45]. Molecular adjuvants, such as CL264, specifically interact with Toll-like receptor 7 (TLR7) to recognize antigens and regulate immune cells, thereby acting as immunomodulators. Compared to the pEGFP-TgIST group, the pEGFP-TgIST + CL264 immunization group showed a prolonged survival time, indicating effective cellular and humoral immune responses after the stimulation of CL264 [46].

## 5. Conclusions

In this study, we successfully developed a novel DNA vaccine pEGFP-TgIST for the first time, and evaluated its immunogenicity and protective efficacy in a murine model. Our results demonstrate that the mice immunized with pEGFP-TgIST exhibited enhanced humoral and cellular immune responses, leading to prolonged survival after infection with the *T. gondii* strain. Moreover, the use of the molecular adjuvant CL264 further improved the immunoprotection, resulting in extended survival times for immunized mice compared to the DNA vaccine group alone. These findings suggest that pEGFP-TgIST holds great promise as a potential DNA vaccine against toxoplasmosis. Considering that many parameters hinder the evaluation of the TgIST DNA vaccine, further investigations can focus on thoroughly evaluating the vaccine using different immunization protocols to optimize its protective efficacy.

## Figures and Tables

**Figure 1 vaccines-12-00577-f001:**
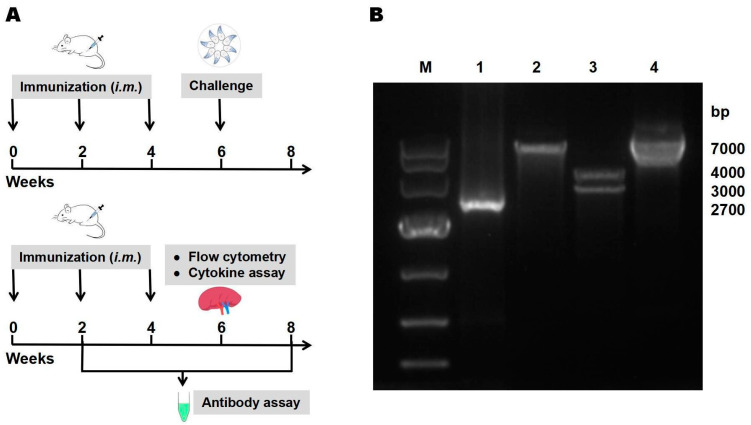
Construction of the novel DNA vaccine pEGFP-TgIST and immunization. (**A**) Flow chart of mice immunization and immunological analyses. (**B**) Gel electrophoresis analysis of pEGFP-TgIST. Lane M: DNA molecular Marker DL 10,000. Lane 1: PCR product of TgIST (approximately 2700 pb). Lane 2: pEGFP-TgIST digested by *Nde* I (approximately 7000 pb). Lane 3: pEGFP-TgIST digested by *Nde* I and *BamH* I into 2 segments (approximately 3000 bp and 4000 pb). Lane 4: pEGFP-TgIST (approximately 7000 pb).

**Figure 2 vaccines-12-00577-f002:**
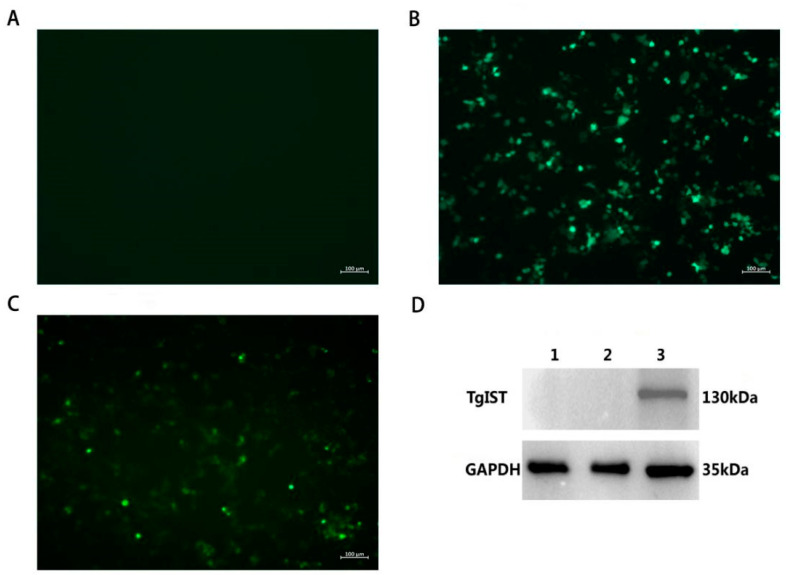
Identification of TgIST expression in vitro by fluorescence microscopic detection and Western blot. (**A**) PBS-treated HEK 293T cells. (**B**) HEK 293T cells transfected with pEGFP-N1; (**C**) HEK 293T cells transfected with pEGFP-TgIST; (**D**) Western blot analysis of TgIST protein. GAPDH serves as a loading control. Lane 1, PBS-treated HEK 293T cells. Lane 2, HEK 293T cells transfected with pEGFP-N1. Lane 3, HEK 293T cells transfected with pEGFP-TgIST.

**Figure 3 vaccines-12-00577-f003:**
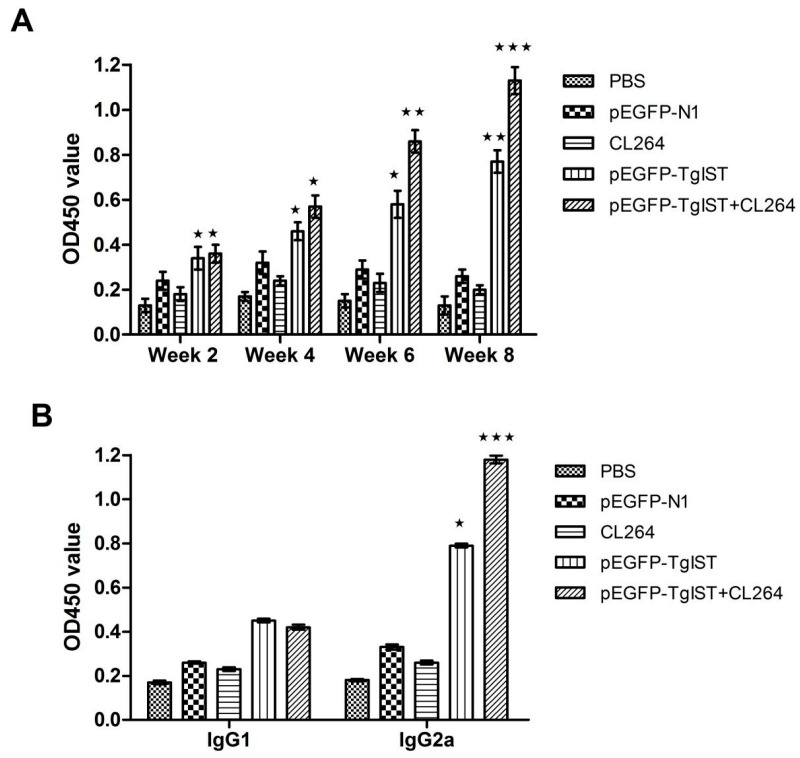
Determination of specific anti-*Toxoplasma* antibodies in sera of immunized BALB/c mice. (**A**) Total IgG antibodies in sera of BALB/c mice on weeks 2, 4, 6 and 8 of the first immunization were determined by ELISA. (**B**) Distributions of IgG subtypes (IgG1 and IgG2a) in sera of BALB/c mice 4 weeks after the final immunization were determined by ELISA. The results are expressed as means ± SD, and statistically significant differences (*p* < 0.05, *p* < 0.01, *p* < 0.001) are indicated by (*), (**) or (***), respectively.

**Figure 4 vaccines-12-00577-f004:**
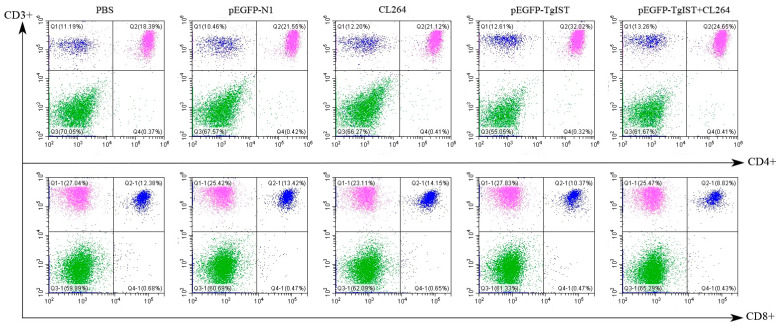
Percentages of CD4^+^ and CD8^+^ T cells subsets in spleens of BALB/c mice (*n* = 3 animals per group) analyzed by flow cytometry.

**Figure 5 vaccines-12-00577-f005:**
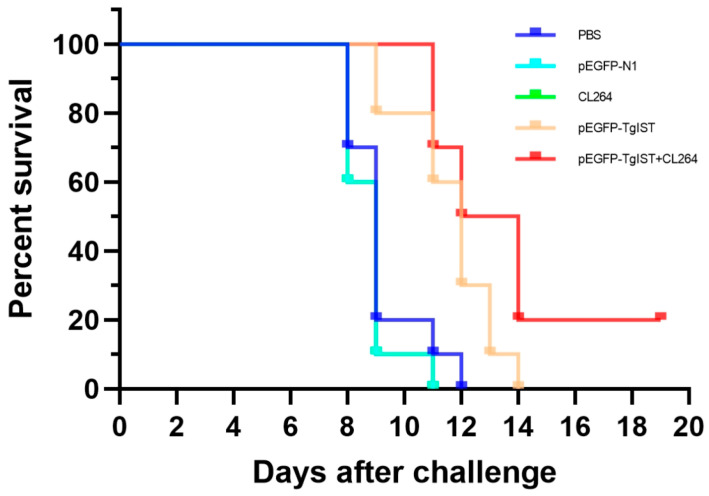
Survival curves of immunized BALB/c mice after challenge with *T. gondii*. The mice (10 per group) were intraperitoneally infected with 100 tachyzoites of the virulent *T. gondii* RH strain 2 weeks after the last immunization. Survival times were monitored daily after the challenge. Overall, survival was expressed as a percentage of surviving animals. Note: The CL264 curve overlaps with the PBS curve and fails to show up in the graph.

**Table 1 vaccines-12-00577-t001:** Immunization groups in this study.

Group	Administration	Immunization Protocol
PBS	Control	100 μL 1 × PBS
pEGFP-N1	Control	100 μg pEGFP-N1 in 100 μL
CL264	Control	40 μg CL264 in 100 μL
pEGFP-TgIST	Vaccination	100 μg pEGFP-TgIST in 100 μL
pEGFP-TgIST + CL264	Vaccination	100 μg pEGFP-TgIST plus 40 μg CL264 in 100 μL

**Table 2 vaccines-12-00577-t002:** Cytokine production by splenocytes of mice after stimulation with *T. gondii* soluble antigen.

Group	Cytokine Concentration (pg/mL)		
	IFN-γ	IL-4	IL-10
PBS	175.24 ± 22.63	35.58 ± 4.22	38.30 ± 3.12
pEGFP-N1	189.16 ± 32.81	41.31 ± 4.47	40.08 ± 3.89
CL264	163.82 ± 17.85	29.96 ± 3.01	36.69 ± 2.92
pEGFP-TgIST	1084.38 ± 194.23 *	35.13 ± 3.60	42.72 ± 4.65
PEGFP-TgIST + CL264	2163.14 ± 331.66 **	30.00 ± 3.46	41.92 ± 4.82

Note: Splenocytes obtained from mice 4 weeks after the final immunization were examined for cytokine production by ELISA. The values for IFN-γ were obtained at 96 h, while the values for IL-10 were obtained at 72 h, and the values for IL-4 were obtained at 24 h. The data are presented as the means ± SD. * *p* < 0.05, ** *p* < 0.01 (immunized vs. control groups).

## Data Availability

The datasets supporting the findings of this study are available within the article and its Appendix A.

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
