# Peer review of "Protective Efficacy of a Novel DNA Vaccine with a CL264 Molecular Adjuvant against *Toxoplasma gondii* in a Murine Model"

_vaccines, 2024, doi:10.3390/vaccines12060577_

Round 1

Reviewer 1 Report

Comments and Suggestions for Authors

Overall is a good try for this zoonosis, I have few suggestions for the authors:

1. Suggest title change. Change IST either using full name or delete it.

2. Line 92 ddH20, please use full name.

3. The main concern is Figure1. What's the size of pGEFP? 4000bp? It has no obvious differences between Lane 1 & 2, and Lane 4, after recombinant plasmid  digested with 2 enzymes, lane 1& 2 also show 2 bands, it maybe smear, but better re-run the gel with low voltage and longer time, it will get better image, right now the quality of this picture is unacceptable.

Comments on the Quality of English Language

Quality of English is good, but some first time using abbreviation needs to be expanded!

Author Response

Overall is a good try for this zoonosis, I have few suggestions for the authors:

  1. Suggest title change. Change IST either using full name or delete it.

Response: Thank you for your suggestion. We have removed the “IST” from the title to eliminate the need for a lengthy explanation in the revised manuscript.

  1. Line 92 ddH20, please use full name.

Response: Thank you for pointing out our negligence. We have included the full name "double-distilled water (ddH2O)" in the revised manuscript.

  1. The main concern is Figure 1. What's the size of pGEFP? 4000bp? It has no obvious differences between Lane 1 & 2, and Lane 4, after recombinant plasmid digested with 2 enzymes, lane 1& 2 also show 2 bands, it maybe smear, but better re-run the gel with low voltage and longer time, it will get better image, right now the quality of this picture is unacceptable.

Response: Thank you for your comments. We have addressed your concerns by repeating the gel electrophoresis to improve the image quality. The sizes of pEGFP, pEGFP-TgIST, and TgIST are 4733bp, 7396bp, and 2673bp, respectively. In Figure 1 of the revised version, Lane M and Lanes 1-4 depict the gel electrophoresis analysis as follows: Lane M shows the DNA molecular Marker DL 10000, Lanes 1-4 show the PCR product of TgIST (2673bp), pEGFP-TgIST digested by Nde I (7396bp), pEGFP-TgIST digested by Nde I and BamH I (approximately 3100bp and 4300bp), and pEGFP-TgIST (7396bp), respectively.

  1. Comments on the Quality of English Language.

Quality of English is good, but some first time using abbreviation needs to be expanded!

Response: Thank you for your comments. We have carefully reviewed the manuscript to ensure that all abbreviations are expanded upon their first mention.

Reviewer 2 Report

Comments and Suggestions for Authors

This is  an interesting study on DNA vaccination against Toxoplasma gondii  targeting TgIST and also making use of a molecular adjuvant and can be published after the following major revisions.

In Figure 3 we can see that CL264 exerts its positive effect starting only from week n. 4. The authors should comment on it in view of a possible utilization of the pEGFP-TgIST-CL264 with respect to only pEGFP-TgIST. Please cite other literature examples in which the adjuvant effects is observed only starting from a similar time.

Moreover, CL264 a 9-benzyl-8 hydroxyadenine glycine, is not described. Thus, at least some details on this derivative at molecular level should be given very briefly to let the reader know what its nature is.

The structure of TgIST protein should be briefly described (may be in the revised Introduction or Discussion) citing the work

  • DOI: https://doi.org/10.1038/s41467-022-31720-7

The authors should at least say that it comprises a core region with internal repeats, which has been identified as essential for its function in blocking STAT1-mediated gene expression. This core region adopts a helical conformation upon binding to phosphorylated STAT1 dimers.

IFN-gamma but not IL-4 and IL-10 levels are increased with the presented DNA vaccination. Please better explain what we can conclude from the lack of any changes in IL-4 and IL-10 also citing the appropriate literature on it

The authors should briefly introduce (in the revised Introduction or Discussion) the importance of DNA vaccination in general including COVID-19, neurodegeneration etc citing at least 

  • DOI https://doi.org/10.3390/vaccines11111706  and https://doi.org/10.1038/s41467-024-44830-1 

I would expect more informative conclusions reporting more details also from a quantitative point of view. 

Authors should provide a better quality figure 4 as the writings and numbers are not readable.

Comments on the Quality of English Language

English language is generally fine

Author Response

This is an interesting study on DNA vaccination against Toxoplasma gondii targeting TgIST and also making use of a molecular adjuvant and can be published after the following major revisions.

Response: Thank you for taking the time to review our manuscript. We are grateful for your insights and believe that following your suggestions will significantly enhance the quality of the paper, making it suitable for publication.

In Figure 3 we can see that CL264 exerts its positive effect starting only from week n. 4. The authors should comment on it in view of a possible utilization of the pEGFP-TgIST-CL264 with respect to only pEGFP-TgIST. Please cite other literature examples in which the adjuvant effects is observed only starting from a similar time.

Response: Thank you for your thorough review and insightful comments on our manuscript. Regarding Figure 3, we observed that CL264 started demonstrating its positive effect from the fourth week. This observation prompts consideration of the potential utility of pEGFP-TgIST-CL264 compared to only pEGFP-TgIST. To support this discussion, we conducted a literature search and found a relevant study (DOI: 10.1016/j.ijbiomac.2023.127228) where specific IgG levels in sera were measured two weeks after the second booster. This study also reported a significant increase in total IgG levels in the recombinant plasmid-adjuvant group after the third immunization, suggesting a distinct timing of adjuvant effects from the recombinant plasmid's effects. We appreciate your suggestion and have included these references to support our discussion on the observed timing of adjuvant effects.

Reference:

Khorshidvand Z, Shirian S, Amiri H, Zamani A, Maghsood AH. Immunomodulatory chitosan nanoparticles for Toxoplasma gondii infection: Novel application of chitosan in complex propranolol-hydrochloride as an adjuvant in vaccine delivery. Int J Biol Macromol. 2023 Dec 31;253(Pt 8):127228. doi: 10.1016/j.ijbiomac.2023.127228. Epub 2023 Oct 14. PMID: 37839605.

Moreover, CL264 a 9-benzyl-8 hydroxyadenine glycine, is not described. Thus, at least some details on this derivative at molecular level should be given very briefly to let the reader know what its nature is.

Response: We appreciate your suggestion and have supplemented the features at molecular level of CL264 in the revised version as “a 9-benzyl-8 hydroxyadenine derivative containing a glycine on the benzyl group” (Page 2, lines 75-76).

The structure of TgIST protein should be briefly described (may be in the revised Introduction or Discussion) citing the work

DOI: https://doi.org/10.1038/s41467-022-31720-7

The authors should at least say that it comprises a core region with internal repeats, which has been identified as essential for its function in blocking STAT1-mediated gene expression. This core region adopts a helical conformation upon binding to phosphorylated STAT1 dimers.

Response: Thank you for your constructive suggestion. We have described the structure of TgIST protein and cited this reference in the revised Introduction as “TgIST comprises a core region with internal repeats, which forms a helical conformation upon binding to the STAT1 dimers and is necessary and sufficient to block STAT1-mediated gene expression” (Page 2, lines 61-63).

IFN-gamma but not IL-4 and IL-10 levels are increased with the presented DNA vaccination. Please better explain what we can conclude from the lack of any changes in IL-4 and IL-10 also citing the appropriate literature on it.

Response: Thank you for your comment. Upon reviewing the literature, we found consistent results showing no significant difference in IL-4 and IL-10 levels between the experimental and control groups. This indicates that the DNA vaccine primarily induces a Th1 response rather than a Th2 response. We have included citations to support this conclusion in the revised version (Page 10, lines 338).

Reference:

Zhu Y, Xu Y, Hong L, Zhou C, Chen J. Immunization With a DNA Vaccine Encoding the Toxoplasma gondii' s GRA39 Prolongs Survival and Reduce Brain Cyst Formation in a Murine Model. Front Microbiol. 2021;12:630682; doi: 10.3389/fmicb.2021.630682.

The authors should briefly introduce (in the revised Introduction or Discussion) the importance of DNA vaccination in general including COVID-19, neurodegeneration etc citing at least

DOI https://doi.org/10.3390/vaccines11111706 

and https://doi.org/10.1038/s41467-024-44830-1

Response: Thank you for your comment. We have introduced the importance and application of DNA vaccines in the revised Discussion as “DNA vaccines have played a protective role in epidemic diseases including the COVID-19 epidemics against SARS-CoV and SARS-CoV-2 variants, and have even been used to potentially prevent or modify the course of neurological diseases” (Page 10, lines 310-313).

I would expect more informative conclusions reporting more details also from a quantitative point of view.

Response: Thank you for your comment. We acknowledge the importance of providing informative conclusions with detailed quantitative analysis, thus we will prioritize addressing these aspects in our future studies.

Authors should provide a better quality figure 4 as the writings and numbers are not readable.

Response: Thank you for your comment. The quality of the Figure 4 in the word file is somewhat impaired, and we have uploaded a high resolution image of Figure 4 to ensure clear readability of the writings and numbers.

Comments on the Quality of English Language. English language is generally fine.

Response: We appreciate your assessment and will ensure that the language remains consistent in the revised version.

Reviewer 3 Report

Comments and Suggestions for Authors

The study describes a study aiming to develop a new vaccine candidate targeting the parasite T. gondii. 

The methods are adequately described and the research design appear to be accurate and well-designed.

I just have a few queries for the authors:

Q1: Fluorescent microscopy: did you get also DAPI and merged confocal fluorescent images of cells?

Q2: WB analysis: the images shown in Fig 2-blots 1 and 2 are two different membranes?

Q3: it appears that about 3 mice survived after immunization and infection with the virulent Tg RH strain in the pEGFP- 258 TgIST+CL264 group. Is it correct? Whereas in all the other groups, including the pEGFP-TgIT group, no mouse survived. So, an adjuvant might play an essential role. Could youplease further better discuss this issue?

Author Response

The study describes a study aiming to develop a new vaccine candidate targeting the parasite T. gondii

The methods are adequately described and the research design appear to be accurate and well-designed.

Response: Thank you for your time to reviewing our manuscript. We appreciate your insights, and we will improve the quality of the manuscript based on your suggestions.

I just have a few queries for the authors:

Q1: Fluorescent microscopy: did you get also DAPI and merged confocal fluorescent images of cells?

Response: Thank you for your inquiry. We did not acquire DAPI or merged confocal fluorescent images of cells. The green fluorescence was directly observed under the fluorescence microscope after EGFP protein expression from plasmid pEGFP-N1 and recombinant plasmid pEGFP-TgIST, respectively.

Q2: WB analysis: the images shown in Fig 2-blots 1 and 2 are two different membranes?

Response: Thank you for your comment. The Western blot images presented in Fig 2D are from the same PVDF membrane but have been divided into two parts for clarity in presentation. The original image is provided in the "Supplementary Materials - Original Images".

Q3: it appears that about 3 mice survived after immunization and infection with the virulent Tg RH strain in the pEGFP- 258 TgIST+CL264 group. Is it correct? Whereas in all the other groups, including the pEGFP-TgIT group, no mouse survived. So, an adjuvant might play an essential role. Could you please further better discuss this issue?

Response: Thank you for bringing this to our attention. It is correct that 2 mice survived after immunization and infection with the RH strain, representing 20% of the 10 mice per group. We apologize for the mistake in indicating "12" mice per group as most mice succumbed within 12 days. This has been rectified in the revised manuscript.

Moreover, we conducted a thorough literature search on CL264 and have gained confidence in its ability to stimulate effective cellular and humoral immune responses, thereby prolonging the survival time of mice. This additional information has been included in the revised Discussion section.

Reference:

Li C, Zhang X, Chen Q, Zhang J, Li W, Hu H, Zhao X, Qiao M, Chen D. Synthetic Polymeric Mixed Micelles Targeting Lymph Nodes Trigger Enhanced Cellular and Humoral Immune Responses. ACS Appl Mater Interfaces. 2018 Jan 24;10(3):2874-2889. doi: 10.1021/acsami.7b14004. Epub 2018 Jan 10. PMID: 29285934.

Round 2

Reviewer 2 Report

Comments and Suggestions for Authors

The manuscript can be accepted for publication in the current form

Comments on the Quality of English Language

English is generally fine

Author Response

Thank you for your recognition and kind help.